# Shape memory polymer resonators as highly sensitive uncooled infrared detectors

Ulas Adiyan[1], Tom Larsen[2], Juan José Zárate [1], Luis Guillermo Villanueva [2] & Herbert Shea [1]*

Uncooled infrared detectors have enabled the rapid growth of thermal imaging applications. These detectors are predominantly bolometers, reading out a pixel's temperature change due to infrared radiation as a resistance change. Another uncooled sensing method is to transduce the infrared radiation into the frequency shift of a mechanical resonator. We present here highly sensitive resonant infrared sensors, based on thermo-responsive shape memory polymers. By exploiting the phase-change polymer as transduction mechanism, our approach provides 2 orders of magnitude improvement of the temperature coefficient of frequency. Noise equivalent temperature difference of 22 mK in vacuum and 112 mK in air are obtained using f/2 optics. The noise equivalent temperature difference is further improved to 6 mK in vacuum by using high-Q silicon nitride membranes as substrates for the shape memory polymers. This high performance in air eliminates the need for vacuum packaging, paving a path towards flexible non-hermetically sealed infrared sensors.

---

[1] Soft Transducers Laboratory (LMTS), École Polytechnique Fédérale de Lausanne (EPFL), 2000 Neuchâtel, Switzerland. [2] Advanced NEMS Group, École Polytechnique Fédérale de Lausanne (EPFL), 1015 Lausanne, Switzerland. *email: herbert.shea@epfl.ch

nfraRed (IR) imaging is rapidly being adopted in a broad range of fields including thermography, medical diagnostics, firefighting, autonomous driving, food inspection and security[1–6]. The two major IR detector technologies are photon detectors (typically cooled) and thermal detectors (typically uncooled). Photon detectors rely on the generation of electron/hole pairs within the detector material when exposed to IR radiation. They are the fastest and the most sensitive IR detection technology; however, they require operation at cryogenic temperatures to minimize thermally generated electron-hole pairs. Cooling systems increase the overall system size and cost. In contrast, thermal detectors transduce IR radiation by measuring a temperature-dependent physical property such as electrical resistance in bolometers[7], electrical polarization in pyroelectric detectors[8], electrical voltage in thermopiles[9], and displacement in thermomechanical detectors[10]. They do not require active cooling, and can be small and power efficient[11]. However, they lag behind photon detectors in terms of sensitivity and response time. Resonant IR sensors[12–19], whose mechanical resonance frequency is temperature dependent, could be a breakthrough for thermal detectors by providing extremely high sensitivity owing to the low noise associated with frequency measurements[20].

Noise Equivalent Temperature Difference (NETD) is the key figure of merit for IR sensor sensitivity. The NETD is the minimum distinguishable temperature change at the radiation source (e.g., IR target, not on the detector). Equation (1) provides one definition[10] of the NETD. $\Delta T_{\mathrm{T}}$ is the temperature change at the target (the IR radiation source). SNR is the signal to noise ratio of the infrared detector output, where $V_{\mathrm{s}}$ is the detector signal level (e.g., readout voltage or output frequency) and $V_{\mathrm{N}}$ is the total noise (e.g., RMS noise voltage, or phase noise) within the system bandwidth[21].

$$\mathrm{NETD} = \frac{\Delta T_{\mathrm{T}}}{\mathrm{SNR}} = \frac{\Delta T_{\mathrm{T}}}{V_{\mathrm{s}}/V_{\mathrm{N}}} \quad (1)$$

For resonant IR sensors[14] Eq. (1) can be rewritten as

$$\mathrm{NETD} = \frac{\Delta T_{\mathrm{T}}}{P_{\mathrm{inc}}} \frac{P_{\mathrm{inc}}}{P_{\mathrm{abs}}} \frac{P_{\mathrm{abs}}}{\Delta T_{\mathrm{D}}} \frac{\sigma_{\mathrm{A}}}{\mathrm{TCF}} \quad (2)$$

$P_{\mathrm{inc}}$ is the power incident on the detector, $P_{\mathrm{abs}}$ is the power absorbed by the detector, $\Delta T_{\mathrm{D}}$ is the temperature change at the detector, $\sigma_{\mathrm{A}}$ is the Allan deviation[22] (a measure for the noise in the frequency), and TCF is the temperature coefficient of the resonance frequency. To enhance sensitivity, one wishes to minimize each ratio in Eq. (2). $(\frac{\Delta T_{\mathrm{T}}}{P_{\mathrm{inc}}})$, is related to the IR optics and is beyond the scope of this paper. $(\frac{P_{\mathrm{inc}}}{P_{\mathrm{abs}}})$ and $(\frac{P_{\mathrm{abs}}}{\Delta T_{\mathrm{D}}})$, are the inverse of the absorbance and the thermal conductance of the sensor (including conduction to the air if not vacuum-packaged). Maximizing the absorbance[23] and minimizing the thermal losses[17,24] of the sensing area improve the sensitivity. Finally, the last term $(\frac{\sigma_{\mathrm{A}}}{\mathrm{TCF}})$ consists of the two key parameters that are specific to resonant IR sensors. High sensitivity can be achieved by increasing the TCF[25,26] and by improving the frequency stability[27–29].

Shape Memory Polymers (SMPs)[30–32] can be used to increase the TCF because their Young's modulus have exceptionally high temperature dependence. SMPs are programmable phase-change materials that can memorize a permanent shape, be deformed and fixed in a temporary shape under given conditions, and later, upon external stimulus, recover their original permanent shape[33–37]. The mechanical properties of SMPs can be changed using stimuli such as temperature[38], light[39], solvent[40] and pressure[41]. The IR sensor technology we report here is based on thermo-responsive SMPs, where the mechanical properties change with the temperature change. In our sensor, the IR radiation heats the SMP, thus changing its mechanical properties. The SMP has a phase change from a rigid state below its glass transition temperature ($T_{\mathrm{glass}}$) to a rubbery state above $T_{\mathrm{glass}}$.

Here we report a highly sensitive resonant IR sensor based on thermo-responsive Shape Memory Polymers (SMPs). We present, to our knowledge, the first use of thermal phase transition property of a polymer as a transduction mechanism for IR detection. Our SMP resonators provide unprecedented responsivity owing to the highest TCF ever reported as an IR sensor. The SMP material itself not only provides the IR to frequency transduction mechanism but is also a good absorber in the long-wave IR (LWIR) range (i.e., from 7 to 14 μm), avoiding the need for an additional absorber layer. The SMP material has low thermal conductivity, enabling good thermal isolation from the environment. These characteristics of the SMP resonators enable highly sensitive IR detection not only in vacuum but also at atmospheric pressure. The sensitivity for IR sensing is further improved by using SMP with high-Q silicon nitride (SiNx) membranes as bimorph resonators.

## Results

**Working principle**. Figure 1 summarizes the working principle of the resonant IR detection technology, based on SMP resonators.

The incident IR radiation from a target induces a temperature change on the SMP resonator which depends on the IR optics, as well as on the absorbance and the thermal coupling of the sensor to the environment. This temperature change leads to a change in the Young's modulus of the SMP material, shifting the mechanical resonance frequency of the resonator. The shift in the resonance frequency can be detected using a frequency readout scheme[42], and then be converted back to a temperature change.

For a given temperature change, the change in the resonance frequency is given by the TCF of the sensor. The TCF can be written as $\mathrm{TCF} = \frac{1}{f_{\mathrm{Res}}} \frac{\partial f_{\mathrm{Res}}}{\partial T_{\mathrm{D}}}$ [SI unit: K$^{-1}$], where $f_{\mathrm{Res}}$ is the resonance frequency. The larger the TCF, the larger the change of the resonance frequency for a given temperature change. For our material, the TCF is temperature dependent, and has in principle a maximum near the glass transition temperature, $T_{\mathrm{glass}}$. We thus aim to operate the sensors heated to near $T_{\mathrm{glass}}$ to achieve the highest TCF.

**SMP material characterization**. Supplementary Fig. 1 shows the dynamic mechanical analysis (DMA) measurement for the commercial MM4520 SMP material, which has a nominal $T_{\mathrm{glass}}$ of 45 °C. The Young's modulus ($E$) is plotted vs. temperature from 20 °C to 80 °C. $E$ drops from 1700 MPa to 10 MPa in this range. We plot the thermal coefficient of the Young's modulus $\mathrm{TCE} = \frac{1}{E} \frac{\partial E}{\partial T_{\mathrm{D}}}$, which has a maximum value of around $\mathrm{TCE}_{\mathrm{peak}} = 0.2\,\mathrm{K}^{-1}$, and is located in the glass transition region at around 50 °C. For low stress disk shape mechanical resonators, the TCF of the resonator can be calculated as half of the TCE of the material, because the mechanical resonance frequency of the resonator is proportional to $\sqrt{E}$. Therefore, the $\mathrm{TCE}_{\mathrm{peak}}$ value implies a $\mathrm{TCF}_{\mathrm{peak}}$ of 0.1 K$^{-1}$ (i.e., 10%). This TCF value will result in a highly sensitive resonant IR sensor if the resonance frequency can be tracked accurately, which is linked to the Q-factor of the resonator. A challenge for the SMP materials is a low Q-factor near $T_{\mathrm{glass}}$. The intrinsic quality factor ($Q_{\mathrm{E}}$) is the inverse of the loss factor of the material. $Q_{\mathrm{E}} = |\tan(\delta)|^{-1} = E'/E''$, where $E'$ is the storage modulus and $E''$ is the loss modulus that were obtained from the DMA measurements. $Q_{\mathrm{E}}$ is plotted vs. temperature in Supplementary Fig. 1. $Q_{\mathrm{E}}$ drops substantially around

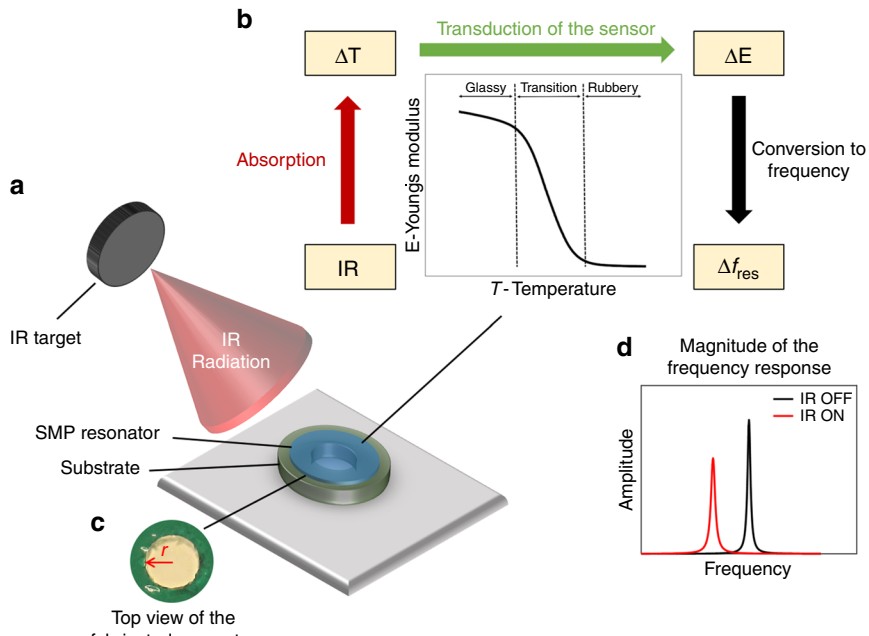

**Fig. 1** Overview for the IR SMP resonant sensor and the working principle. **a** IR sensor system illustration. **b** The working principle of the IR SMP sensor: The incident IR radiation from the IR target causes a temperature change, $\Delta T$, on the SMP resonator. This leads to a change in the Young's modulus, $\Delta E$, of the SMP material, which shifts the resonance frequency, $\Delta f_{res}$, of the SMP resonator. **c** Top view of the fabricated SMP resonator. **d** Schematic frequency response of the IR SMP resonator when the IR illumination is on and off: the change in the IR target's temperature can be detected from the change in resonance frequency

the glass transition temperature. There is a trade-off between the Q-factor and the TCE or TCF, since the maximum loss for the SMP material coincides with the TCE$_{peak}$ due to the high viscoelasticity around $T_{glass}$. Operating the IR SMP resonator away from $T_{glass}$ results in higher Q-factor, but lower TCF. Consequently, the optimum operation temperature may not be exactly at $T_{glass}$.

Another approach to the trade-off between Q-factor and TCF is to create a bimorph consisting of a layer of SMP material bonded on a high-Q mechanical resonator. The bimorph has a higher Q-factor than bare SMP, but a lower TCF because most high-Q materials have TCF orders of magnitude smaller than SMP. To study this, in addition to bare SMP resonators, we also used highly stressed (HS) silicon nitride (SiNx) resonators with Q-factors over 10,000 at room temperature in high vacuum.

**SMP resonator characterization sequence**. We fabricated circular and square shaped resonators as these geometries are easier to fabricate and provide large IR radiation absorber area than bridge type structure. The dimensions of the circular shaped resonators are 520 μm in radius and 10 μm in thickness and the dimensions of the square shaped resonators are 1 mm × 1 mm × 150 nm, unless anything else is stated. The resonator is placed onto a Polymethyl methacrylate (PMMA) substrate having a thickness of 1 mm, and with a central hole to define the resonator shape. Fabrication and assembly of the SMP resonators are described in the Methods section.

The SMP resonator is mounted on a piezo-disk actuator and placed on a heater with a proportional–integral–derivative (PID) temperature controller system, which controls the operation temperature of the resonator ($T_{sub}$), using a thermistor placed on the substrate (Supplementary Fig. 2). The details on the PID controller can be found in the Methods section. The temperature is increased from $T_{sub} = 25$ °C to $T_{sub} = 50$ °C in steps of 5 °C. The change in resonance frequency and Q-factor are monitored

by measuring the displacement vs. frequency curves using a Laser Doppler Vibrometer (LDV). TCF and the frequency stability noise ($\sigma_A$) of this resonator are determined for every operation temperature. Finally, the NETD and response time are measured.

**Resonance frequency and Q**. Figure 2a shows the resonance frequency of the 1st flexural mode with respect to temperature in vacuum (~$10^{-3}$ Pa). For each $T_{sub}$, the measured displacement vs. frequency responses of the SMP membrane is fitted to a Lorentzian function to determine the resonance frequency and the Q-factor. Supplementary Fig. 3 shows the frequency response of the SMP membrane for 2 operation temperatures ($T_{sub} = 25$ °C in the glassy state, $T_{sub} = 45$ °C in the transition state). The resonance frequency vs. temperature measurements are compared with analytical values computed[42] from the Young's modulus ($E$) vs. temperature data obtained from the DMA measurements (Supplementary Fig. 1). The experimental data agrees well with the analytical solution (Fig. 2a). The resonance frequency scales as $f_{Res} \sim \sqrt{E(T)}$.

Figure 2b plots the Q-factor of the SMP membrane in vacuum vs. temperature ($T_{sub}$). Both the measured Q and the intrinsic material quality factor $Q_E$ from the DMA results agree well. Q has its highest value near room temperature (in glassy state, Q ≈ 43), it drops down to Q ≈ 1–2 in the transition region. While there are many possible energy dissipation factors such as air damping and anchor losses[43], the dominant energy loss mechanism for our case is the intrinsic material loss.

**TCF, frequency stability, and ultimate sensitivity**. From the resonance frequency of the SMP resonator as a function of temperature, it is straightforward to calculate the TCF: $\text{TCF} = \frac{1}{f_{Res}} \frac{\Delta f_{Res}}{\Delta T_{sub}}$. From the TCF and frequency stability noise, characterized by the Allan deviation (AD)-$\sigma_A$, one can calculate the minimum detectable change in the substrate temperature: $\delta T_{sub} = \frac{\sigma_A}{\text{TCF}}$.

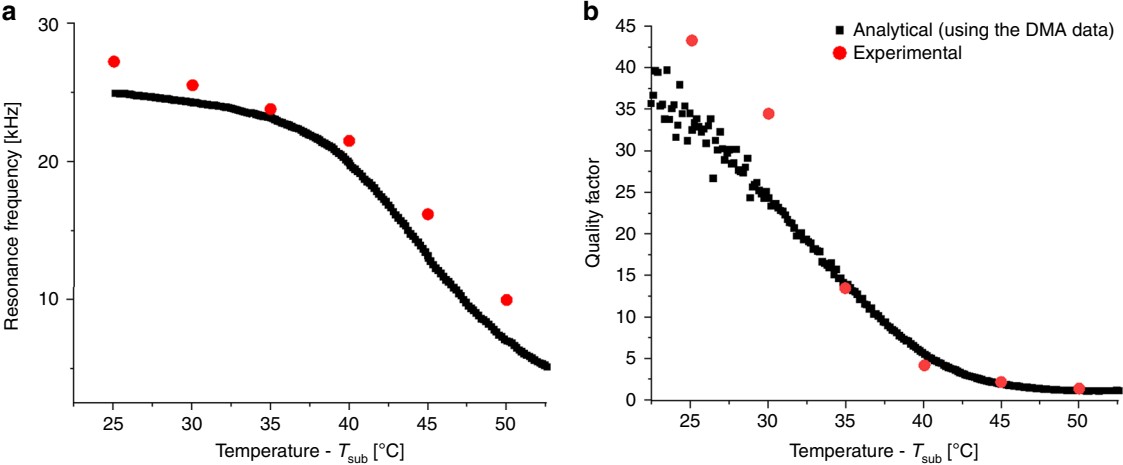

**Fig. 2** SMP resonator characterization. **a** Resonance frequency and **b** Q-factor of the SMP resonator vs. temperature. Both plots show measured data in vacuum and the analytical solution based on DMA data

Figure 3a shows the AD measurements of a SMP resonator as a function of integration time ($\tau$) for substrate temperatures between 25–50 °C. The minimum AD values ($\sigma_A \approx 8.6 \times 10^{-7}$ for $\tau = 460$ ms) occur at a substrate temperature of $T_{sub} = 30$ °C. These AD values are 2 orders of magnitude higher than typical MEMS resonators ($\sigma_A \approx 10^{-8.2}$), deduced from data from the literature[22] for devices with similar masses (~10 ng). As we increased the operation temperature, the frequency instability increases. This is probably due to the drop in the Q-factor, as the AD is inversely related to the Q-factor of the resonator[22]. The amplitude of the output signal (resonator displacement) is another important parameter that improves the frequency stability[44]. We operated the resonator in all cases close to the maximum displacement limit of our LDV instrument. Therefore, as we increase the operation temperature, we also increase the piezo driving voltage to compensate for the reduction in Q.

The maximum frequency stability occurs for an integration time between $\tau = 200$ ms and $\tau = 400$ ms depending on the operation temperature. The frequency instability increases for longer integration times, probably due to thermal drift. For the temperature detection sensitivity analysis, we used the AD values corresponding to an integration time of $\tau = 500$ ms, as this matches the sensor thermal time constant. (See Supplementary Note 1 for scaling the SMP diameter down to 52 µm resonator for a computed time constant of 7 ms).

Figure 3b shows the measured TCF and Allan Deviation values for an integration time of $\tau = 500$ ms. The TCF are calculated according to the normalized frequency change within each 5 °C step using the measurements from Fig. 2a. The best TCF is ~8% $K^{-1}$ at an operation temperature of $T_{sub} = 45$ °C. This TCF is 2 orders of magnitude higher than the best TCF reported so far (~0.1% $K^{-1}$) for IR resonant sensors[25]. In addition, the TCF is more than an order of magnitude higher than resonant temperature sensors[45–47], for which the best reported TCF is 1.7% $K^{-1}$.

For each data point in Fig. 3b, the minimum detectable temperature change on the substrate ($\delta T_{sub}$), which corresponds to direct temperature detection sensitivity, is calculated. The best temperature detection sensitivity is 63 µK at $T_{sub} = 30$ °C operation temperature, as shown in Fig. 3c. This value implies that the simple SMP resonant sensor architecture offers temperature detection sensitivity comparable or better than the performances of the state-of-the-art resonant temperature sensors[45,46] or photonic temperature sensors with far more complex structures[48,49].

**NETD measurements**. The experimental setup for the characterization of the IR SMP sensors is illustrated in Supplementary Fig. 4. IR radiation from a target at controlled temperature is focused onto the SMP resonator. The absorbed IR radiation creates a temperature change on the IR sensor, $\Delta T_D$, which is much smaller than the change in temperature at the target, $\Delta T_T$. $\frac{\Delta T_D}{\Delta T_T}$ represents the detector to target temperature ratio, which is a function of the IR optics, detector area, absorbance of the IR sensor and thermal conductance (See Supplementary Note 2 for a detailed analysis).

The absorbance of a 10 µm thick SMP sample was measured using Fourier transform infrared (FTIR) spectrometer throughout the LWIR spectrum (Supplementary Fig. 5). A 10 µm, 23 µm, and 57 µm thick SMP film absorbs 48, 69 and 84% of the IR radiation in the 7–14 µm spectral range (See Methods for the absorbance measurements), respectively. Since the SMP material is a good absorber in LWIR range, there is no need for additional absorber layer, and 10 µm thick layer is sufficient for this study.

To test SMP resonators as IR sensors, the IR radiation emitted from a black resistive heater was modulated using a rotating chopper (see Methods for the details). Figure 4a shows the IR response of the resonator, for two different temperature at the IR source. This leads to two different temperature differences of the source with respect to the ambient, which we call the blackbody temperature difference ($\Delta T_{bb}$). The temperature for the resonator substrate is held at $T_{sub} = 30$ °C (since it is the optimal operation temperature for the device), and the resonator is operated in vacuum (~$10^{-3}$ Pa). The temperature changes of $\Delta T_{bb} = 5$ °C and $\Delta T_{bb} = 15$ °C at the IR source result in a frequency shift of $\Delta f_1 \approx 5$ Hz and $\Delta f_2 \approx 15$ Hz, respectively. To determine the NETD of the IR sensor, we use NETD = $\sigma_A / \text{TCF}_{bb}$ where the TCF$_{bb}$ is the TCF of our sensor with respect to the 1 °C blackbody temperature difference and can be expressed as $\text{TCF}_{bb} = \frac{1}{f_{Res}}\frac{\partial f_{Res}}{\partial T_{bb}}$. We measure the NETD as low as 22 mK.

The thermal time constant is also deduced as 376 ± 8 ms based on the modulated IR response (Supplementary Fig. 6), which is an important figure of merit to evaluate the response time and the potential of the sensor for real time video applications (see Supplementary Note 1).

Figure 4b shows the IR response of the resonator, in vacuum (~$10^{-3}$ Pa), as well as in atmospheric pressure at the operation temperature of $T_{sub} = 30$ °C. A temperature change of $\Delta T_{bb} = 15$ °C at the target results in a frequency change of $\Delta f_{air} \approx 5.4$ Hz

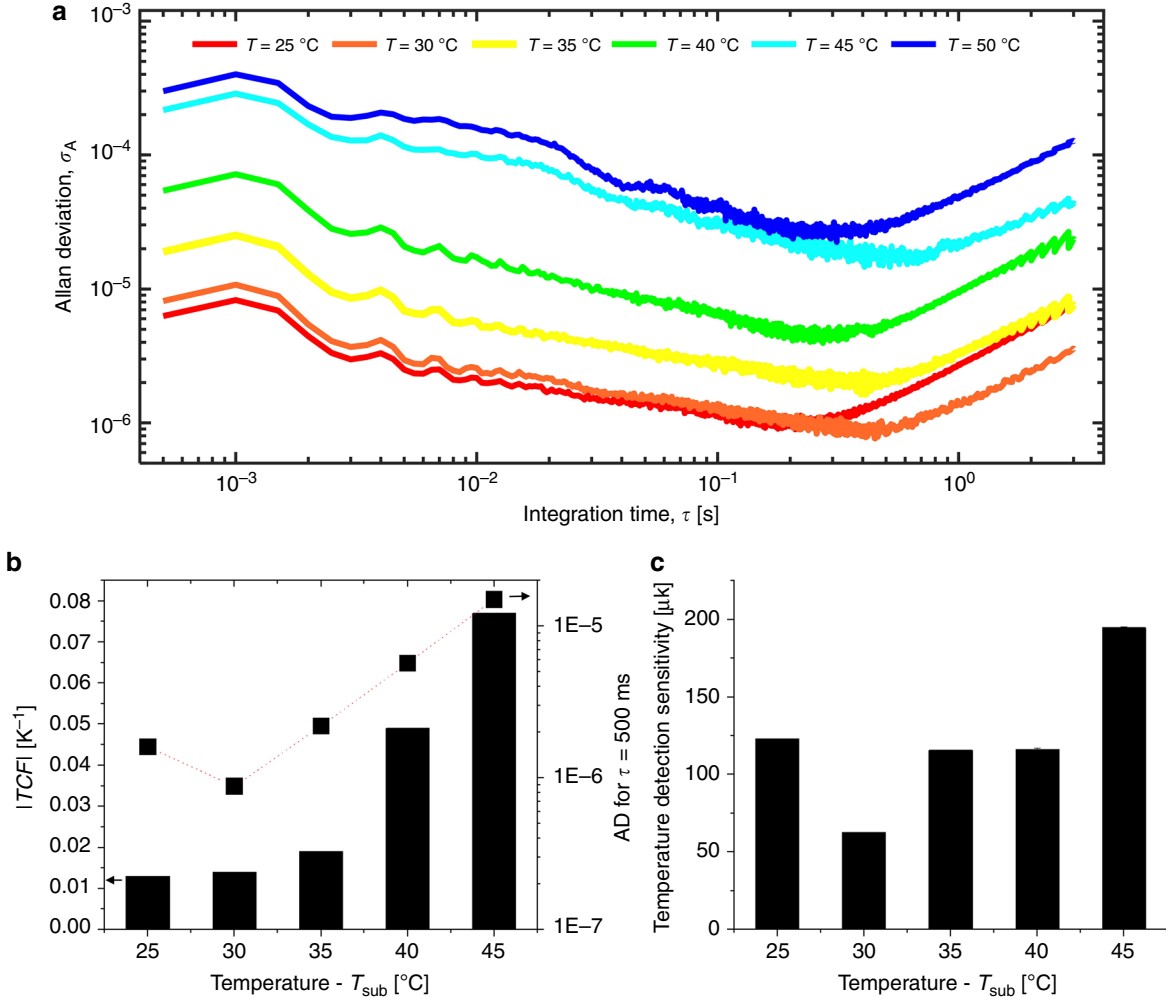

**Fig. 3** TCF, frequency stability and sensitivity analysis for the SMP resonator. **a** Allan Deviation measurements as a function of integration time for different operation temperatures obtained in vacuum (~$10^{-3}$ Pa). **b** TCF (left $y$-axis) and AD measurement (right $y$-axis) for an integration time of $\tau = 500$ ms vs. operation temperature ($T_{sub}$). **c** Computed temperature detection sensitivity ($\delta T_{sub} = \frac{\sigma_A}{TCF}$) vs. operation temperatures. The best derived sensitivity is 63 μK at $T_{sub} = 30$ °C operation temperature

in air as compared to $\Delta f_{vacuum} \approx 15$ Hz in vacuum. The thermal conductance from the membrane to the substrate through the air degrades the detector to target temperature ratio (See Supplementary Note 2). This can be seen in Fig. 4b, where the thermal time constant of the membrane in air is lower than in vacuum due to heat conduction through the air. In order to determine the NETD in air, similar AD measurements are performed at atmospheric pressure. The NETD at $T_{sub} = 30$ °C in air pressure is calculated as 162 mK. Even though NETD in air is ~7 times higher than in vacuum, it is nevertheless comparable to the NETD from the state-of-the-art resonant IR sensors that operate in vacuum[14,15].

Finally we measure the NETD for operation temperatures between 25–45 °C in vacuum and air (Fig. 4c). At 25 and at 30 °C, the NETD is less than 45 mK (best NETD of 22 mK) in vacuum and 160 mK (best NETD of 112 mK) in air. We compare the measured NETDs with the predicted values in Supplementary Note 3 and see good agreement between them (Supplementary Fig. 7).

**Improvement in Q-factor using silicon nitride membranes.** We have fabricated devices consisting of 15 μm of the SMP material bonded on high-stress (HS, around 1 GPa) silicon nitride (SiNx) membranes, 1 mm × 1 mm × 150 nm, which have Q-factors

greater than 10000 in vacuum. These composite resonators have a higher Q-factor and a lower TCF than bare SMP resonators. The resonance frequency of the SiNx/SMP resonator is measured as 147637 ∓ 6 Hz and 144573 ∓ 6 Hz with Q-factors of 737 ∓ 8 and 1050 ∓ 4, at $T_{sub} = 25$ °C and $T_{sub} = 30$ °C, respectively. For more details see Supplementary Fig. 8. The TCF is ~0.4% at $T_{sub} = 30$ °C.

Figure 5a shows the IR response of a SiNx/SMP membrane, for $\Delta T_{bb} = 5$ °C at the IR source. The temperature of the substrate is held at $T_{sub} = 30$ °C and the resonator is operated in vacuum (~$10^{-3}$ Pa). A frequency shift of $\Delta f_3 \approx 15$ Hz is measured.

Figure 5b shows the AD measurements of the SiNx/SMP membrane at $T_{sub} = 30$ °C. The frequency instability improves an order of magnitude compared to SMP resonators due to the improved Q-factor. For an integration time of $\tau = 500$ ms, the measured *NETD* is 6 mK. The thermal time constant is also determined as 311 ± 1 ms based on the modulated IR response. These same experiments were repeated with smaller SiNx/SMP square membranes, of dimension 500 × 500 μm. Supplementary Fig. 9 shows the test results from smaller membrane. The NETD is calculated as 38 mK for the integration time of $\tau = 250$ ms, with a thermal time constant of 225 ± 6 ms. Table 1 compares the performance of the IR SMP and SMP/SiNx resonant sensors of different sizes and for different pressure conditions.

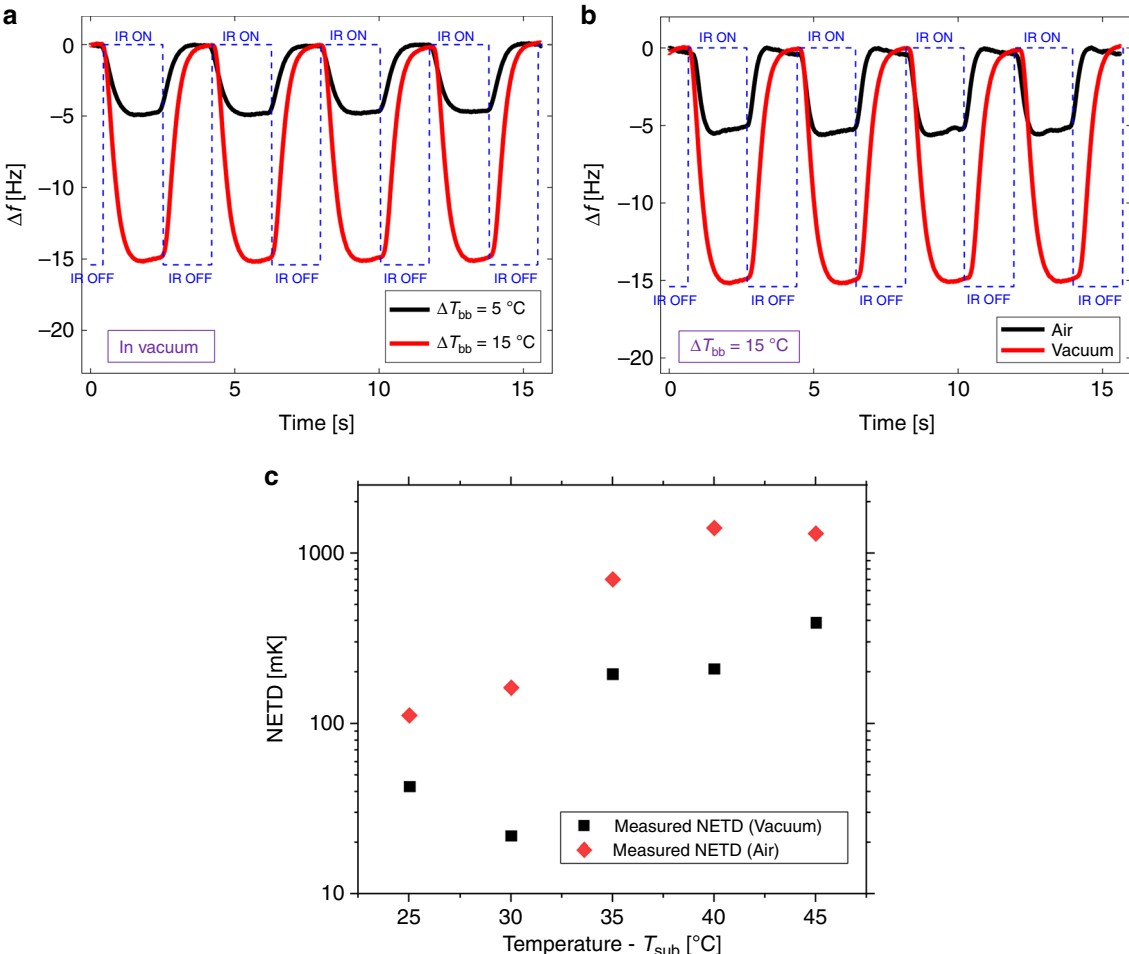

**Fig. 4** NETD measurements of the IR SMP sensor. **a** The frequency response of the SMP IR sensor in vacuum at $T_{sub} = 30\,°C$ operation temperature when the target temperature is periodically changed by $\Delta T_{bb} = 5°$ and by $\Delta T_{bb} = 15°$. **b** The frequency response of the IR resonator at $T_{sub} = 30\,°C$ operation temperature in vacuum and at 1 atmosphere. **c** Measured NETD vs. operation temperature in vacuum and in air

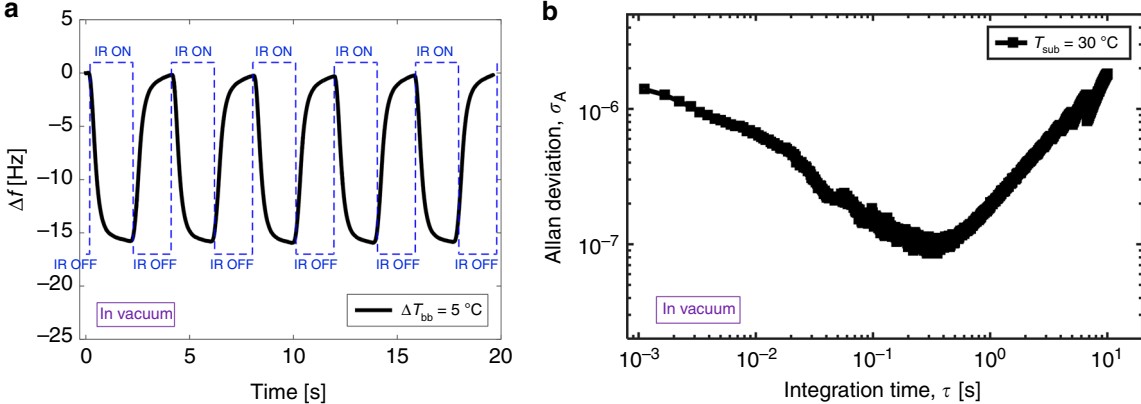

**Fig. 5** NETD measurements of the SiNx/SMP IR sensor. **a** The frequency response of the composite SiNx/SMP IR sensor in vacuum at $T_{sub} = 30\,°C$ operation temperature when the target temperature is periodically changed by $\Delta T_{bb} = 5°$. **b** Allan Deviation measurements as a function of integration time at $T_{sub} = 30\,°C$ for the SiNx/SMP resonant IR sensor in vacuum ($\sim 10^{-3}$ Pa)

## Discussion

We report the highest TCF reported for resonant IR detectors, with a direct temperature detection sensitivity (63 μK with SMP resonators and 28 μK with SiNx/SMP resonators) comparable to or better than state-of-the-art temperature sensors. For IR sensing,

we measure an NETD as low as 22 mK (in vacuum) and 112 mK (in air) using an optical system having an f-number of 2 (F# = 2) for the bare SMP resonator having a radius of 520 μm. We measure even lower NETD (6 mK) by using composite SiNx/SMP resonators. Compared to bare SMP resonators, we obtain 30 times

**Table 1 Performance comparison of SMP and SiNx/SMP IR resonant sensors: measured TCF, NETD and time constant for SMP and SiNx/SMP resonant sensors of different sizes and shapes and in vacuum or ambient atmosphere conditions**

| Sensor | Size | Vacuum condition | TCF [%K$^{-1}$] | NETD [mK] | Time constant [ms] |
|---|---|---|---|---|---|
| SMP circular resonator | radius = 520 μm, $t$ = 10 μm | High vacuum (~10$^{-3}$ Pa) | 1.2 | 22 | 376 ± 8 |
| SMP circular resonator | radius = 520 μm, $t$ = 10 μm | Air (1 atm) | 1 | 160 | 210 ± 10 |
| SiNx/SMP square resonator | length = 1 mm, $t$ = 150 nm | High vacuum (~10$^{-3}$ Pa) | 0.4 | 6 | 311 ± 1 |
| SiNx/SMP square resonator | length = 500 μm, $t$ = 150 nm | High vacuum (~10$^{-3}$ Pa) | 0.5 | 38 | 225 ± 6 |

All results are measured at $T_{sub}$ = 30 °C

increase in Q-factor (from Q = 34 to Q = 1050) for the SiNx/SMP resonators, which leads to 10 times improvement in frequency stability at $T_{sub}$ = 30 °C. However, the TCF of the composite resonator drops by 3× (from 1.2 to 0.4%) at $T_{sub}$ = 30 °C. The SiNx/SMP resonators lead to improvement in the sensitivity of approximately 3×.

It is important to note that we can improve the IR radiation collection efficiency by decreasing the f-number of the IR optics. We could improve the sensitivity 4 times by changing the optics to F# = 1. There is room to improve the absorption by a factor of two (e.g., by forming a resonant cavity). With these improvements, the NETD would drop to 2.75 mK for SMP resonator and to the sub-mK levels for SiNx/SMP resonators (For the theoretical NETD analysis, see Supplementary Figs. 10–11; Supplementary Note 2).

The NETD of the IR sensor is closely related with the response time of the sensor. Higher response times lead to IR detection with higher sensitivity. We developed a thermal model to analyze the response time of the sensor (Supplementary Figs. 12–14; Supplementary Note 1). The thermal time constant ($\tau_{th}$) of the SMP resonator defines the response time of the sensor, which can be expressed as $\tau_{th} = \frac{C_{total}}{G_{total}}$, where $G_{total}$ is the total thermal conductance and $C_{total}$ is the heat capacity of the SMP resonator. The response time of the sensor is proportional to the heat capacity and to the inverse of the thermal conductance of the SMP membrane. Our SMP resonators have a thermal time constant of 376 ± 8 ms in vacuum and 210 ± 10 ms in atmospheric pressure.

To decrease the response time, one must decrease the radius and/or the thickness of the membrane, while considering the effect of both parameters on sensitivity via the thermal conductance and absorber area. For instance, 10 times decrease in the radius (from 520 μm to 52 μm) reduced the thermal time constant by nearly 100× in vacuum, down to 7 ms.

We demonstrated the first use of a phase-change polymer as the transduction mechanism of a highly sensitive IR resonating sensor. The SMP material itself provides the transduction mechanism, as well as is a good absorber in the LWIR range (7–14 μm). Our SMP sensors can operate in atmospheric pressure with a very small degradation compared to vacuum operation, due to the high sensitivity of the SMP sensor and to intrinsic high thermal isolation. SMPs can be patterned using established MEMS processes to achieve 50 μm or smaller pixel sizes in large arrays. Capacitive transduction can be used for actuation and for readout. We envision that arrays of SMP or SiNx/SMP resonators can be used as uncooled THz detectors thanks to the high TCF (associated with the material property) and low noise (associated with frequency measurements).

## Methods

**Fabrication**. The fabrication process of the SMP membrane is shown in Supplementary Fig. 15. For the devices reported in this work we used MM4520 SMP pellets from SMP Technologies. In order to produce a castable SMP solution, the SMP pellets are dissolved in DMF at a weight ratio of 1:5 and mixed overnight at 80 °C. The fabrication starts with (1) cleaning an A4-size polyethylene terephthalate (PET) sheet, which is used as the substrate for the SMP membrane, and

(2) blade casting of 100 nm thick sacrificial Teflon™ AF amorphous fluoroplastic layer *(from Chemours Company)* on this PET sheet to help the release of the SMP membrane. (3) SMP solution is casted on the sacrificial layer using *Zehntner ZAA2300 film applicator coater*. Each casting step is preceded by an oxygen plasma treatment. After casting, there is a curing step at 80 °C on a hotplate for ~3 h. The process continues with (4) patterning the SMP membrane using a laser cutter (*Trotec Speedy 300*). (5) a PMMA plate with 1 mm thickness is used to form the substrate for the resonator and an adhesive layer (*ARclear 8932EE, Adhesive Research*) is applied on this PMMA plate. (6) CNC (*CNC Basic 540 from Step-Four*) is used to pattern the PMMA to form circular holes for the resonator. After having the patterned PMMA substrate and the SMP membrane, (7) they are assembled together. Finally, (8) laser cutter is used to cut out single resonator units.

Silicon nitride membranes are fabricated by deposition of either low or high stress silicon nitride (SiNx) via LPCVD on a double-side polished wafer. Openings in the backside of the nitride are made using photolithography and plasma etching. A through wafer KOH etching is then used to release the membranes. A 15 μm thick SMP is coated on the membranes using a spin-coater to form SiNx/SMP membranes.

**Absorber characterization**. Fourier Transform Infrared (FTIR) absorption measurements is performed with THERMO Nicolet 8700 FTIR on bare SMP samples. The spectrum of the transmittance, $T$, is measured. The absorption spectrum, $A$, is then directly extracted from the following expression[50]: $A = 1-T-R$, assuming the total reflectivity ($R$) from the sample in the LWIR is around 5%[51].

**Measurement**. Supplementary Fig. 4 shows the experimental setup for the IR detection. The SMP resonators are bonded on piezo-disk actuators (*Noliac—NAC 2014, Thorlabs—PA4FEW*), using a thermal adhesive (*Fischer Elektronik—WLFT 404 23 × 23*). The SMP resonator on the piezo-disk actuator is fixed on a Peltier element (*Multicomp—MCPE-127-10-25*) using the same thermal adhesive. A commercial PID temperature controller system (*Meerstetter Engineering GmbH—TEC-1091*) controls and adjusts the operation temperature of the resonators with a PT-100 temperature sensor which is located on the PMMA substrate. In order to select the parameter sets of the PID controller, the auto-tuning process of the PID controller's service software is used for the optimization of the system.

The magnitude and the phase of the frequency response of the SMP resonators are measured using a laser Doppler vibrometer (*Polytec—MSV 400*). The samples are placed into a vacuum chamber, which has a ZnSe optical window without AR-coating (*Crystran LTD—ZNSEP50-3*) to allow transmission of both and the IR radiation and the red laser light for displacement readout. The visible light has normal incidence on the sensor, while the IR radiation has an oblique incidence around 55°–60°. A collimator setup[52] with two AR-coated IR ZnSe lenses (*Thorlabs —LA7656-G-Ø1"-f = 25.4 mm and LA7542-G-Ø1"-f = 50.1 mm*) are implemented to produce a controllable irradiance that is independent of source-target distance as illustrated in Supplementary Fig. 4. Due to space limitations under the LDV, we use the second lens in the collimator with a focal length of 50.1 mm with 25.4 mm diameter that results in a lower collection efficiency due to the use of a lens with f-number of 2. A resistive heater painted black (*DBK—HP04−1/04-24*) is used as an IR source that is placed in front of a rotating chopper (*Stanford Research Systems, SR540*). The IR target is calibrated with a commercial IR thermometer (*PeakTech 4950*) to define the temperature difference of the target with the ambient.

The modulated IR radiation from this IR target passes through the IR lens system and illuminates the SMP resonator. This creates a temperature change on the resonator, which changes its Young's modulus and hence the resonance frequency, monitored by the LDV. The internal generator of a lock-in amplifier (*Stanford Research Systems, SR550*) is used to drive the piezo-disk actuator at the fixed driving frequency (near resonance), while the output of the LDV is connected to the same lock-in amplifier to track the phase difference between the actuation signal and the sensor output. Thus, the change in the phase signal is proportional to the temperature difference between the chopper blade and the IR target. This phase signal is converted to the frequency using the open-loop phase response of the resonator. The slope of the phase response near the resonance frequency ($m = \Delta\varphi/\Delta f$) is determined, and this slope is used to convert phase to frequency ($f(t) = \varphi(t)/m$). The phase and amplitude data from the lock-in amplifier is collected using an oscilloscope (*Agilent Technologies, MSO9104A*). For the data IR

response measurements (Fig. 4a, b) of bare SMP resonators, a moving average filter ($N = 1001$) is used to smooth the data, which was acquired with an acquisition rate of 25 kHz. The Allan Deviation is measured in an open-loop configuration—constant drive frequency. The resonators are not irradiated with IR during the measurements. We follow the same procedure as in ref. [22] to calculate the Allan deviation.

To characterize the SiNx/SMP resonators, the same experimental setup is used as for SMP resonators. However, we use a different lock-in amplifier (UHFLI, Zurich Instruments), with a larger measurement bandwidth. The frequency tracking is directly measured by means of a phase-locked loop (PLL). The frequency output measurements of SiNx/SMP resonators during IR detection tests and the Allan Deviation measurements are implemented with this PLL system. The remaining experimental procedure and the Allan deviation calculation methods are the same as for the bare SMP resonators. For the data of IR response measurements (Fig. 5a, b, Supplementary Fig. 9a, b), a moving average filter ($N = 101$) is used to smooth the data, which was acquired with an acquisition rate of 1.8 kHz.

## Data availability

The datasets generated during the current study are available from the corresponding author upon reasonable request.

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

## Acknowledgements

We acknowledge financial support from the EPFL. We thank M. Lehmann for Fourier-transform infrared spectroscopy (FTIR) measurements carried out in the PV-Lab of EPFL.

## Author contributions

H.S. and J.J.Z. conceived the concept. U.A designed the devices, developed thermal and mechanical models, fabricated the samples, designed and performed the experiments, analyzed and processed the data. T.L. assisted in the frequency and AD measurements. L. G.V. and H.S. advised on the design of the samples, the experimental methods, and on data interpretation. U.A. wrote the paper. All authors read, edited, and discussed the paper and agree with the claims made in this work. H.S. coordinated and supervised the research.

## Competing interests

The authors declare no competing interests.
