## [Peer Review File · Nature Communications]

Reviewers' comments:

Reviewer #1 (Remarks to the Author):

This paper reports on a highly sensitive resonant IR sensor based on a thermoresponsive shape memory polymer (SMP). The authors claimed that the SMP has very good absorption in IR wavelengths, obviating the need for an absorber layer. Furthermore, their sensor provides 2 orders of magnitude improvement of the temperature coefficient of frequency (TCF) by exploiting the phase transition of the SMP as the transduction mechanism. Overall this work is achieved properly and would provide the necessary thrust for SMPs as sensors. However, it suggests a careful revision of the manuscript with respect to improving the language as well as the presentation aspects. Moreover, some characterizations and experimental details are required to be supplemented.

1. In Figure 4, the response time is about 1 second. Except for the obvious frequency change, the response time is also important for the sensitivity of the sensor. So please explain how to shorten the response time in your design, if any.
2. Please explain whether the thermal capacity of the SMP layer is related to the response time.
3. Please pay attention to some English writing. e.g., In Page 1, line 12. In the sentence (We present here a highly sensitive, simple to fabricate resonant IR sensor...), the phrase of simple to fabricate is used improperly, which should be corrected.

Reviewer #4 (Remarks to the Author):

Adiyan et al presents a highly sensitive uncooled IR detector based on thermo-responsive shape memory polymers. Compared to the traditional IR detectors, the uncooled IR detector can be worked not only in vacuum but also at atmospheric pressure with good performance. This work achieves interesting progress on IR detector. However, there are several issues that need to be clarified before I can recommend publication of this manuscript in this journal.

1. In the characterization of SMP, the author correlates the temperature change with the modular change of SMP. However, the initial stimulus in IR detector is IR light. The author should correlate the intensity of IR light with the modulu change of SMP.
2. In SMP resonator characterization, the author carried out a sample of a radius of 520 um and a thickness of 10 um. What is the penetration ability of IR light to SMP? If the thickness of the sample is too large, the IR light may not irradiate the bottom of the sample and cause the change of modulus. Does it affect the test results? Please examine the effect of sample's thickness on the test results.
3. Why should the sample be disk shape? Does this shape help to improve the test results?
4. Figure 5 is too simple. Simple annotations can be added to the corresponding positions in the diagram.
5. The paper should be checked carefully for the unsuitable spelling/format in the word.

“Shape Memory Polymer Resonators as Highly Sensitive Uncooled Infrared Detectors”, by Ulas Adiyen, Tom Larsen, Juan José Zárate, Luis Guillermo Villanueva and Herbert Shea.

Response to reviewer comments

We used blue text color for the reviewers’ comments, black text color for our responses and red text color to show the modifications to the manuscript.

Comments of Reviewer #1:

General Comment: *This paper reports on a highly sensitive resonant IR sensor based on a thermo-responsive shape memory polymer (SMP). The authors claimed that the SMP has very good absorption in IR wavelengths, obviating the need for an absorber layer. Furthermore, their sensor provides 2 orders of magnitude improvement of the temperature coefficient of frequency (TCF) by exploiting the phase transition of the SMP as the transduction mechanism. Overall this work is achieved properly and would provide the necessary thrust for SMPs as sensors. However, it suggests a careful revision of the manuscript with respect to improving the language as well as the presentation aspects. Moreover, some characterizations and experimental details are required to be supplemented.*

Answer: We thank the reviewers for their constructive feedback on the manuscript. We modified the manuscript according to the suggestions of the reviewers. In addition, we added a completely new subsection to the manuscript named ***“Improvement in Q-factor using silicon nitride membranes”***, which includes detailed experimental characterization of silicon nitride (SiNx) membranes coated with SMP films to achieve high Q-factor resonant IR sensors.

Modifications to the manuscript related to the new subsection and high-Q IR SiNx/SMP resonators:

- A sentence is added to the abstract p.1 line 17:

The NETD is further improved to 6 mK in vacuum by using high-Q silicon nitride membranes as substrates for the SMP.

- A paragraph is added to introduce high-Q IR SiNx/SMP resonators in p.7 line 128:

Another approach to the trade-off between Q-factor and TCF is to create a bimorph consisting of a layer of SMP material bonded on a high-Q mechanical resonator. The bimorph has a higher Q-factor than bare SMP, but a lower TCF because most high-Q materials have TCF orders of magnitude smaller than SMP. To study this, in addition to bare SMP resonators, we also used highly stressed (HS) silicon nitride (SiNx) resonators with Q-factors over 10,000 at room temperature in high vacuum.

- New subsection added p.15 line 271 onwards:

Improvement in Q-factor using silicon nitride membranes. We have fabricated devices consisting of 15 μm of the SMP material bonded on high-stress (HS, around 1 GPa) silicon nitride (SiNx) membranes, 1 mm x 1 mm x 150 nm, which have Q-factors greater than 10000 in vacuum. These composite resonators have a higher Q-factor and a lower TCF than bare SMP resonators. The resonance frequency of the SiNx/SMP resonators are measured as $147637 \pm 6 \text{ Hz}$ and $144573 \pm 6 \text{ Hz}$ with Q-factors of 737 ± 8 and 1050 ± 4 , at $T_{\text{sub}}=25 \text{ }^\circ\text{C}$ and $T_{\text{sub}}=30 \text{ }^\circ\text{C}$ respectively. For more details see Supplementary Figure 7. The TCF is $\sim 0.4\%$ at $T_{\text{sub}}=30 \text{ }^\circ\text{C}$.

Figure 5.a shows the IR response of a SiNx/SMP membrane, for $\Delta T_{\text{bb}} = 5 \text{ }^\circ\text{C}$ at the IR source. The temperature of the substrate is held at $T_{\text{sub}}=30 \text{ }^\circ\text{C}$ and the resonator is operated in vacuum ($\sim 10^{-3} \text{ Pa}$). A frequency shift of $\Delta f_3 \approx 15 \text{ Hz}$ is measured.

Figure 5: a) The frequency response of the composite SiNx/SMP IR sensor in vacuum at $T_{sub}=30$ °C operation temperature when the target temperature is periodically changed by $\Delta T_{bb}=5^\circ$. b) Allan Deviation measurements as a function of integration time at $T_{sub}=30$ °C for the SiNx/SMP resonant IR sensor in vacuum ($\sim 10^{-3}$ Pa).

Figure 5.b shows the AD measurements of the SiNx/SMP membrane at $T_{sub}=30$ °C. The frequency instability improves an order of magnitude compared to SMP resonators due to the improved Q-factor. For an integration time of $\tau=500$ ms, the measured NETD is 6 mK. The thermal time constant is also determined as 311 ± 1 ms based on the modulated IR response. These same experiments were repeated with smaller SiNx/SMP square membranes, of dimension $500 \mu\text{m} \times 500 \mu\text{m}$. Supplementary Figure 8 shows the test results from smaller membrane. The NETD is calculated as 38 mK for the integration time of $\tau=250$ ms, with a thermal time constant of 225 ± 6 ms. Table 1 compares the performance of the IR SMP and SMP/SiNx resonant sensors of different sizes and for different pressure conditions.

Table 1: Measured TCF, NETD and time constant for SMP and SiNx/SMP resonant sensors of different sizes and shapes and in vacuum or ambient atmosphere conditions. All results are measured at $T_{sub}=30$ °C.

Sensor	Size	Vacuum Condition	TCF (%/K)	NETD [mK]	Time Constant [ms]
SMP circular resonator	radius=520 μm , t=10 μm	High Vacuum ($\sim 10^{-3}$ Pa)	1.2	22	376 ± 8
SMP circular resonator	radius=520 μm , t=10 μm	Air (1 atm)	1	160	210 ± 10
SiNx/SMP square resonator	length=1 mm t= 150 nm	High Vacuum ($\sim 10^{-3}$ Pa)	0.4	6	311 ± 1
SiNx/SMP square resonator	length=500 μm t= 150 nm	High Vacuum ($\sim 10^{-3}$ Pa)	0.5	38	225 ± 6

- New Figure: Supplementary Figure 7 in Supplementary File p.4 line 49 onwards:

Supplementary Figure 7: Displacement vs. frequency of a composite silicon nitride (SiNx) & SMP membrane at 25°C and at 30 °C measured by Laser Doppler Vibrometer. The size of the square membrane is 1 mm x 1 mm and the thickness of the SMP is 15 μm .

- New Figure: Supplementary Figure 8 in Supplementary File p.5 line 54 onwards:

Supplementary Figure 8: a-) The frequency response of the smaller SiNx/SMP IR resonant sensor in vacuum at $T_{sub}=30$ °C operation temperature when the target temperature is periodically changed by $\Delta T_{bb} = 5^\circ$. A moving average filter ($N=101$) is used to smooth the data, which was acquired with an acquisition rate of 1.8 kHz. b-) Allan Deviation measurements as a function of integration time at $T_{sub} = 30$ °C for the same IR resonant sensor in vacuum ($\sim 10^{-3}$ Pa). The top view of the fabricated resonator is shown at the bottom left corner of the figure.

- The modifications in the Discussion part added p.17 line 307 onwards:

We measure even lower NETD (6 mK) by using composite SiNx/SMP resonators. Compared to bare SMP resonators, we obtain 30 times increase in Q-factor (from $Q=34$ to $Q=1050$) for the SiNx/SMP resonators, which leads to 10 times improvement in frequency

stability at $T_{sub}=30$ °C. However, the TCF of the composite resonator drops by 3x (from 1.2% to 0.4%) at $T_{sub}=30$ °C. The SiNx/SMP resonators leads to improvement in the sensitivity of approximately 3x.

It is important to note that we can improve the IR radiation collection efficiency by decreasing the f-number of the IR optics. We could improve the sensitivity 4 times by changing the optics to $F\#=1$. There is room to improve the absorption by a factor of two (e.g. by forming a resonant cavity). With these improvements the NETD would drop to 2.75 mK for SMP resonator and to the sub-mK levels for SiNx/SMP resonators.

- The modifications in the Methods related to the fabrication of SiNx/SMP membranes, added p.21 line 367 onwards:

Silicon nitride membranes are fabricated by deposition of either low or high stress silicon nitride (SiNx) via LPCVD on a double-side polished wafer. Openings in the backside of the nitride are made using photolithography and plasma etching. A through wafer KOH etching is then used to release the membranes. A 15 μm thick SMP is coated on the membranes using a spin-coater to form SiNx/SMP membranes.

- The modifications in the Methods related to the measurement of SiNx/SMP membranes, added p.23 line 421 onwards:

To characterize the SiNx/SMP resonators, the same experimental setup is used as for SMP resonators. However, we use a different lock-in amplifier (UHFLI, Zurich Instruments), with a larger measurement bandwidth. The frequency tracking is directly measured by means of a phase-locked loop (PLL). The frequency output measurements of SiNx/SMP resonators during IR detection tests and the Allan Deviation measurements are implemented with this PLL system. The remaining experimental procedure and the Allan deviation calculation methods are the same as for the bare SMP resonators. For the data of IR response measurements (Figure 5.a and b, Supplementary Figure 8.a and b), a moving average filter ($N=101$) is used to smooth the data, which was acquired with an acquisition rate of 1.8 kHz.

Comment 1: *In Figure 4, the response time is about 1 second. Except for the obvious frequency change, the response time is also important for the sensitivity of the sensor. So please explain how to shorten the response time in your design, if any.*

Answer 1: We thank the reviewer for highlighting this point in the manuscript. We agree that the response time, which is related to the thermal time constant of the sensor, is very important for the sensitivity of the sensor. There is a trade-off between the thermal time constant and sensitivity of the sensor based on the thermal conductance of the sensor. On one hand, by increasing the thermal conductance, one can decrease the time constant and the response time. On the other hand, the thermal isolation of the sensor will be worse, which leads to decreased sensitivity.

We developed a thermal model to analyze the response time of the sensor with the sensor geometry. According to our analysis in Supplementary Note 1, in order to decrease the response time, one needs to decrease the radius and/or the thickness of the membrane. For instance, 10 times decrease in the radius of the bare SMP resonators (from 520 μm to 52 μm) makes the thermal time constant almost 100 times lower in vacuum, resulting in 7 ms according to our model, which is sufficient for video applications.

In addition, we also added experimental characterization from the SMP coated silicon nitride square membranes (SiN_x) in order to show the method for decreasing the response time. Measurements in vacuum reveal time constants of 311 ± 1 ms (for 1 mm x 1 mm membrane) and 225 ± 6 ms (for 500 μm x 500 μm membrane). These experimental results show that the decrease in the size of the membrane decreases the time constant, so does the response time.

Following the comment of the reviewer, we added an analysis to the Discussion section of the manuscript based on the developed thermal model and the characterization tests of different size SiN_x /SMP membranes.

Modifications to the manuscript:

- Text added p.18 line 319 onwards:

The NETD of the IR sensor is closely related with the response time of the sensor. Higher response times lead to IR detection with higher sensitivity. We developed a thermal model to analyze the response time of the sensor (Supplementary Note 1). The thermal time constant (τ_{th}) of the SMP resonator defines the response time of the sensor, which can be expressed as

$\tau_{th} = \frac{C_{total}}{G_{total}}$, where G_{total} is the total thermal conductance and C_{total} is the heat capacity of the SMP resonator. The response time of the sensor is proportional to the heat capacity and to the inverse of the thermal conductance of the SMP membrane. Our SMP resonators have a thermal time constant of 376 ± 8 ms in vacuum and 210 ± 10 ms in atmospheric pressure.

To decrease the response time, one must decrease the radius and/or the thickness of the membrane, while considering the effect of both parameters on sensitivity via the thermal conductance and absorber area. For instance, 10 times decrease in the radius (from 520 μm to 52 μm) reduced the thermal time constant by nearly 100x in vacuum, down to 7 ms.

- Text added in the Supplementary File p.11 line 166 onwards to evaluate the thermal time constant of the SiNx/SMP resonators based on the thermal model:

We calculate the thermal time constant of the SiNx/SMP resonators by modifying the thermal model slightly, as the geometry changes from a circular membrane to square membrane and as the thickness of the SMP changes from 10 to 15 μm . The calculated thermal time constant from the thermal model is 876 ms for 1 mm x 1 mm and 240 ms for 500 μm x 500 μm sized SiNx/SMP square membranes in vacuum at $T=30$ °C, whereas the exponential fit from the measurements in vacuum reveal time constants of 311 ± 1 ms (for 1 mm x 1 mm membrane) and 225 ± 6 ms (for 500 μm x 500 μm membrane) respectively (See Figure 5.a of the manuscript and Supplementary Figure8.a). So, the thermal model and experimental results show that the decrease in the size of the membrane decreases the thermal time constant.

Comment 2: *Please explain whether the thermal capacity of the SMP layer is related to the response time.*

Answer 2: We thank the reviewer for this comment. Yes, the response time linearly changes with the thermal or heat capacity (C_{total}) of the SMP layer. This can be seen from the expression for the thermal time constant (τ_{th}) of the SMP membrane: $\tau_{th} = \frac{C_{total}}{G_{total}}$, where G_{total} is the total thermal conductance and C_{total} is the heat capacity of the SMP layer. So, the higher the heat capacity or the lower the thermal conductance, higher the thermal time constant and slower the device.

Following the comment of the reviewer, we added a new paragraph on the relation of heat capacity and thermal conductance with the response time in the Discussion section.

Modifications to the manuscript:

Text added p.18 line 322 onwards:

The thermal time constant (τ_{th}) of the SMP resonator defines the response time of the sensor, which can be expressed as $\tau_{th} = \frac{C_{total}}{G_{total}}$, where G_{total} is the total thermal conductance and C_{total} is the heat capacity of the SMP resonator. The response time of the sensor is proportional to the heat capacity and to the inverse of the thermal conductance of the SMP membrane.

Comment 3: *Please pay attention to some English writing. e.g., In Page 1, line 12. In the sentence (We present here a highly sensitive, simple to fabricate resonant IR sensor...), the phrase of simple to fabricate is used improperly, which should be corrected.*

Answer 3: We reviewed the entire text carefully and rewrote several sections to improve clarity and style.

Comments of Reviewer #4:

General Comment: *Adiyan et al presents a highly sensitive uncooled IR detector based on thermo-responsive shape memory polymers. Compared to the traditional IR detectors, the uncooled IR detector can be worked not only in vacuum but also at atmospheric pressure with good performance. This work achieves interesting progress on IR detector. However, there are several issues that need to be clarified before I can recommend publication of this manuscript in this journal.*

Answer: We would like to thank the reviewers for their constructive feedback on the manuscript. We modified the manuscript according to the suggestions of the reviewers. In addition, we added a completely new subsection to the manuscript named “**Improvement in Q-factor using silicon nitride membranes**”, which includes detailed experimental characterization of silicon nitride membranes coated with shape memory polymers to achieve high Q-factor resonant IR sensors. Please see response to Reviewer #1 above.

Comment 1: *In the characterization of SMP, the author correlates the temperature change with the modular change of SMP. However, the initial stimulus in IR detector is IR light. The author should correlate the intensity of IR light with the modulus change of SMP.*

Answer 1: We thank the reviewer for highlighting this point. As the reviewer pointed out, the initial stimulus in IR detector is the IR radiation. The IR radiation can change the mechanical properties of the shape memory polymer (SMP) material in two different ways. If the SMP is a photo-responsive type (i.e. light-induced shape memory polymers¹), then the IR radiation directly changes the mechanical properties of the SMP material.

However, our IR detector technology is based on the thermo-responsive SMPs (*MM4520 SMP pellets from SMP Technologies*²), where the SMP mechanical properties change only with the temperature (not with illumination or by chemical means). The IR radiation as the initial stimulus indirectly changes the mechanical properties of the thermo-responsive SMP material by changing the temperature of SMP material. Therefore, as it is

¹ Lendlein, Andreas, et al. "Light-induced shape-memory polymers." *Nature* 434.7035 (2005): 879.

² <http://www2.smp techno.com/en/smp/>

very common for thermo-responsive SMPs³, we correlated the Young's modulus change with the temperature.

We added new text to clarify the utilized SMP material and the related transduction mechanism for the IR sensor:

Modification in the manuscript:

Text added p.4 line 64 onwards:

The mechanical properties of SMPs can be changed using stimuli such as temperature³⁸, light³⁹, solvent⁴⁰ and pressure⁴¹. The IR sensor technology we report here is based on thermo-responsive SMPs, where the mechanical properties change with the temperature change. In our sensor, the IR radiation heats the SMP, thus changing its mechanical properties. The SMP has a phase change from a rigid state below its glass transition temperature (T_{glass}) to a rubbery state above T_{glass} .

Added to the References of the Manuscript:

38. Aksoy, B. *et al.* Latchable microfluidic valve arrays based on shape memory polymer actuators. *Lab. Chip* (2019).
39. Lendlein, A., Jiang, H., Jünger, O. & Langer, R. Light-induced shape-memory polymers. *Nature* 434, 879 (2005).
40. Gu, X. & Mather, P. T. Water-triggered shape memory of multiblock thermoplastic polyurethanes (TPUs). *Rsc Adv.* 3, 15783–15791 (2013).
41. Fang, Y. *et al.* Reconfigurable photonic crystals enabled by pressure-responsive shape-memory polymers. *Nat. Commun.* 6, 7416 (2015).

³ Lendlein, Andreas, and Steffen Kelch. "Shape-memory polymers." *Angewandte Chemie International Edition* 41.12 (2002): 2034-2057.

Comment 2: *In SMP resonator characterization, the author carried out a sample of a radius of 520 μm and a thickness of 10 μm . What is the penetration ability of IR light to SMP? If the thickness of the sample is too large, the IR light may not irradiate the bottom of the sample and cause the change of modulus. Does it affect the test results? Please examine the effect of sample's thickness on the test results.*

Answer 2: We thank the reviewer for the helpful comment. In order to assess the penetration ability of the IR light to SMP, we carried out absorbance measurements for samples with different thicknesses using Fourier transform infrared (FTIR) spectrometer throughout the long wave IR spectrum (7-14 μm). We measured 3 different samples with the thicknesses of $t_1=10 \mu\text{m}$, $t_2=23 \mu\text{m}$ and $t_3= 57 \mu\text{m}$. The thicker sample has better absorption, which we plot in Supplementary Figure 5 that provided below.

After analyzing the effect of thickness on the IR absorption, we also analyzed the effect of the thickness on the temperature change at the top and bottom of the SMP membrane, when exposed to IR radiation. Since the thickness of the membrane is much smaller than the length of the membrane, the thermal conductance from the center of the membrane to the substrate is much smaller than the thermal conductance from top to bottom of the membrane. Therefore, the temperature is same at the top and the bottom surface of the SMP membrane due to an absorbed IR radiation.

So, the Young's modulus change with the absorbed IR radiation will be the same at the top and bottom surface, even for a thicker membrane ($\sim t=50 \mu\text{m}$). In order to clarify this point, we modified the Supplementary Figure 9, where we defined a new resistor element (with a thermal conductance of G_3) in the thermal model between the top and bottom surface of the membrane and defined it in the Equation 6 of the Supplementary file to show that it is negligible.

Modification in the manuscript:

- Text added for the effect of the thickness on the absorption of IR radiation in p.12 line 226 onwards and Supplementary Figure 5 is modified for the absorption characterization of SMP for different film thicknesses.

A 10 μm , 23 μm and 57 μm thick SMP film absorbs 48%, 69% and 84% of the IR radiation in the 7-14 μm spectral range (See Methods for the absorbance measurements) respectively. Since the SMP material is a good absorber in LWIR range, there is no need for additional absorber layer, and 10 μm thick layer is sufficient for this study.

Supplementary Figure 5: Absorption spectrum of the 10 μm , 23 μm and 57 μm thick SMP sheets in the wavelength range of 7 μm – 14 μm . A THERMO Nicolet 8700 FTIR spectrometer was used to obtain the IR absorption of the SMP with different thicknesses.

- Supplementary Figure 9 is modified by adding a new resistor element (with a thermal conductance of G_3) to analyze the effect of the thickness on the temperature change due to absorbed radiation in the Supplementary File p.6 line 71 onwards:

Supplementary Figure 9: The thermal model for the SMP resonant IR detector. A heat flux is applied on the membrane and the heat flow from the membrane to the substrate goes through the indicated resistances, which is shown with the equivalent circuit of the thermal model.

- Text added to analyze the effect of the thickness on the temperature change due to absorbed radiation in the Supplementary File p.8 line 102 onwards:

$$G_1 = k_{SMP} \frac{2\pi r t}{r} = 2\pi k_{SMP} t \quad (1)$$

$$G_2 = k_{SMP} \frac{\pi(r_m^2 - r^2)}{t} \quad (2)$$

$$G_3 = k_{SMP} \frac{\pi r^2}{t} \quad (3)$$

$$G_{air} = k_{air} \frac{\pi r^2}{t_{air}} \quad (4)$$

G_1 depends only on the thermal conductivity of the SMP and the thickness of the membrane (Equation (1)). For the given geometry in Supplementary Figure 9, G_1 is very small compared to G_2 and G_3 , thus, G_2 and G_3 are negligible in the calculations. Since G_3 is

negligible as compared to G_1 and G_{air} (e.g. G_3 is 3 orders of magnitude higher than G_1 for $t=50\ \mu\text{m}$, which refers to the resistor from the membrane to the substrate), the temperature is same at the top and the bottom surface of the SMP membrane due to an absorbed IR radiation.

Comment 3: *Why should the sample be disk shape? Does this shape help to improve the test results?*

Answer 3: We thank the reviewer for bringing up this point. The resonance frequency of the SMP resonators change with the temperature change, which is based on the change of the Young's modulus ($f_{Res} \sim \sqrt{E}$). This Young's modulus change is an intrinsic material property, which is independent from the shape of the resonator. Therefore, to first order and more most modes, the frequency change of the resonators does not depend on the shape. Thus, we selected a symmetrical and easy to fabricate shape (i.e. circular disk shape) for the characterization tests, compared to bridge structures, membranes provide a large IR radiation absorber fill factor. In addition, we added a completely new subsection to the manuscript named "Improvement in Q-factor using silicon nitride membranes" that includes characterization results of SMP coated silicon nitride membranes. In this section, we achieved high sensitive IR detection using square disk shape membranes.

Thus, even though we used two resonator geometries (circular and square disk shape), both of them achieved highly sensitive IR sensing, because the transduction mechanism of the sensor depends on the material property rather than the resonator shape.

We added two sentences to the "*SMP Resonator Characterization*" section to clarify the reasoning for the selection on the shape of the resonator:

Modification in the manuscript:

- Text added p.7 line 134 onwards:

We fabricated circular and square shaped suspended resonators as these geometries are easier to fabricate and provide large IR radiation absorber area than bridge type structure.

Comment 4: *Figure 5 is too simple. Simple annotations can be added to the corresponding positions in the diagram.*

Answer 4: We thank the reviewer for the helpful comment. We modified the Figure 5 (after revisions, it is Figure 6) and added descriptions on the figure for each step.

Modification in the manuscript:

- We modified Figure 6 (previously Figure 5) in p.21 line 363 onwards:

Figure 6: Schematic illustration of the fabrication process of the bare SMP resonators. The steps for preparing the solution and casting of the SMP is similar to the one used in ref. 36. The resonator thickness is 10 μm .

Comment 5: *The paper should be checked carefully for the unsuitable spelling/format in the word.*

Answer 5: We reviewed the entire text carefully, checked the spelling and format, and rewrote many sections to improve clarity.

REVIEWERS' COMMENTS:

Reviewer #1 (Remarks to the Author):

I suggest acceptance of the revised manuscript at the present format since all my concerned questions have been settled down.

Reviewer #4 (Remarks to the Author):

This paper presents a shape memory polymer resonators as highly sensitive uncooled infrared detectors, the detector showed 2 orders of magnitude improvement of temperature coefficient of frequency. The NETD is 22mK in vacuum and can be further improved to 6mK in vacuum by using high-Q silicon nitride membranes as substrates for the SMP. This work should be accepted after referring to following issues.

1. Page 7, Paragraph 3, line 2, PID should be clarified
2. Page 9, Paragraph 2, Line 2, "the minimum AD value" should be added the detailed data of it. Similarly, Line 4 "typical MEMs resonators from the literature", the detailed data should be provided directly.
3. Page 12, paragraph 1, " The best TCF reported so far for IR resonant sensors" should be added the detailed data directly. Similarly, "The TCF is more than an order of magnitude higher when compared to the resonant temperature sensor", the detailed data should be proved directly.

“Shape Memory Polymer Resonators as Highly Sensitive Uncooled Infrared Detectors”, by Ulas Adiyani, Tom Larsen, Juan José Zárate, Luis Guillermo Villanueva and Herbert Shea.

Response to reviewer comments

We used blue text color for the reviewers’ comments, black text color for our responses, and red text color to show the modifications to the manuscript.

Comments of Reviewer #4:

General Comment: *This paper presents a shape memory polymer resonators as highly sensitive uncooled infrared detectors, the detector showed 2 orders of magnitude improvement of temperature coefficient of frequency. The NETD is 22mK in vacuum and can be further improved to 6mK in vacuum by using high-Q silicon nitride membranes as substrates for the SMP. This work should be accepted after referring to following issues.*

Answer: We thank the reviewers for their constructive feedback on the manuscript. We modified the manuscript accordingly.

Comment 1: *Page 7, Paragraph 3, line 2, PID should be clarified.*

Answer 1: We made modifications in the main text and in the Methods section in order to clarify the utilized PID controller.

Modifications to the manuscript:

- Text added p.8 line 3 onwards:

The SMP resonator is mounted on a piezo-disk actuator and placed on a heater with a proportional–integral–derivative (PID) temperature controller system, which controls the operation temperature of the resonator (T_{sub}), using a thermistor placed on the substrate (Supplementary Figure 2). The details on the PID controller can be found in the Methods section.

- Text added p.21 line 10 onwards:

A commercial PID temperature controller system (*Meerstetter Engineering GmbH - TEC-1091*) controls and adjusts the operation temperature of the resonators with a PT-100 temperature sensor which is located on the PMMA substrate. In order to select the parameter sets of the PID controller, the auto-tuning process of the PID controller's service software is used for the optimization of the system.

Comment 2: *Page 9, Paragraph 2, Line 2, "the minimum AD value" should be added the detailed data of it. Similarly, Line 4 "typical MEMs resonators from the literature", the detailed data should be provided directly.*

Answer 2: We thank the reviewer for this comment. We provided the related data on the AD values in the manuscript as can be seen below.

Modifications to the manuscript:

Text added p.10 line 3 onwards:

The minimum AD values ($\sigma_A \approx 8.6 \times 10^{-7}$ for $\tau = 460$ ms) occur at a substrate temperature of $T_{\text{sub}}=30$ °C. These AD values are 2 orders of magnitude higher than typical MEMS resonators ($\sigma_A \approx 10^{-8.2}$), deduced from data from the literature²² for devices with similar masses (~ 10 ng).

Comment 3: *Page 12, paragraph 1, "The best TCF reported so far for IR resonant sensors" should be added the detailed data directly. Similarly, "The TCF is more than an order of magnitude higher when compared to the resonant temperature sensor", the detailed data should be proved directly.*

Answer 3: We thank the reviewer for this comment. We provided the related data on the TCF values in the manuscript.

Modifications to the manuscript:

Text added p.11 line 11 onwards:

The best TCF is $\sim 8\% \text{ K}^{-1}$ at an operation temperature of $T_{\text{sub}}=45 \text{ }^\circ\text{C}$. This TCF is 2 orders of magnitude higher than the best TCF reported so far ($\sim 0.1\% \text{ K}^{-1}$) for IR resonant sensors²⁵. Additionally, the TCF is more than an order of magnitude higher than resonant temperature sensors⁴⁵⁻⁴⁷, for which the best reported TCF is $1.7\% \text{ K}^{-1}$.